# Reflexion: Language Agents with Verbal Reinforcement Learning

**Noah Shinn**
Northeastern University
noahshinn024@gmail.com

**Federico Cassano**
Northeastern University
cassano.f@northeastern.edu

**Ashwin Gopinath**
MIT
agopi@mit.edu

**Karthik Narasimhan**
Princeton University
karthikn@princeton.edu

**Shunyu Yao**
Princeton University
shunyuy@princeton.edu

## Abstract

Large language models (LLMs) have been increasingly used to interact with external environments (e.g., games, compilers, APIs) as goal-driven agents. However, it remains challenging for these language agents to quickly and efficiently learn from trial-and-error as traditional reinforcement learning methods require extensive training samples and expensive model fine-tuning. We propose *Reflexion*, a novel framework to reinforce language agents not by updating weights, but instead through linguistic feedback. Concretely, Reflexion agents verbally reflect on task feedback signals, then maintain their own reflective text in an episodic memory buffer to induce better decision-making in subsequent trials. Reflexion is flexible enough to incorporate various types (scalar values or free-form language) and sources (external or internally simulated) of feedback signals, and obtains significant improvements over a baseline agent across diverse tasks (sequential decision-making, coding, language reasoning). For example, Reflexion achieves a 91% pass@1 accuracy on the HumanEval coding benchmark, surpassing the previous state-of-the-art GPT-4 that achieves 80%. We also conduct ablation and analysis studies using different feedback signals, feedback incorporation methods, and agent types, and provide insights into how they affect performance. We release all code, demos, and datasets at `https://github.com/noahshinn024/reflexion`.

## 1 Introduction

Recent works such as ReAct [30], SayCan [1], Toolformer [22], HuggingGPT [23], generative agents [19], and WebGPT [17] have demonstrated the feasibility of autonomous decision-making agents that are built on top of a large language model (LLM) core. These methods use LLMs to generate text and 'actions' that can be used in API calls and executed in an environment. Since they rely on massive models with an enormous number of parameters, such approaches have been so far limited to using in-context examples as a way of teaching the agents, since more traditional optimization schemes like reinforcement learning with gradient descent require substantial amounts of compute and time.

In this paper, we propose an alternative approach called *Reflexion* that uses verbal reinforcement to help agents learn from prior failings. Reflexion converts binary or scalar feedback from the environment into verbal feedback in the form of a textual summary, which is then added as additional context for the LLM agent in the next episode. This self-reflective feedback acts as a 'semantic' gradient signal by providing the agent with a concrete direction to improve upon, helping it learn from prior mistakes to perform better on the task. This is akin to how humans iteratively learn to

37th Conference on Neural Information Processing Systems (NeurIPS 2023).

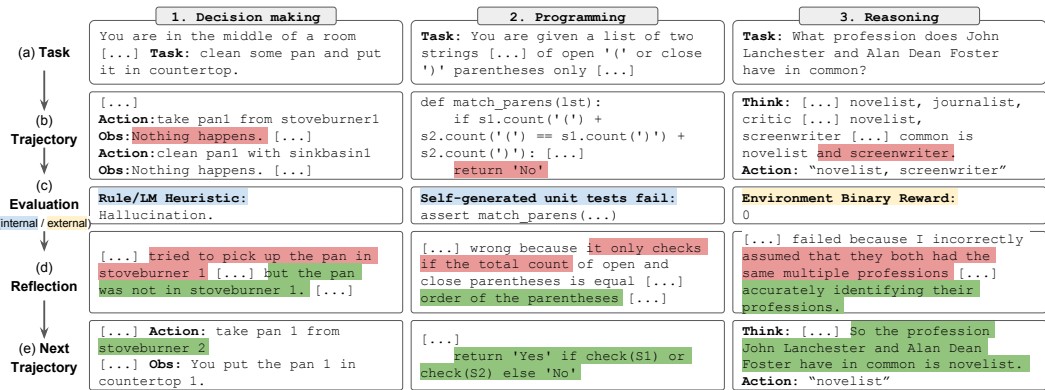

Figure 1: Reflexion works on decision-making 4.1, programming 4.3, and reasoning 4.2 tasks.

accomplish complex tasks in a few-shot manner – by reflecting on their previous failures in order to form an improved plan of attack for the next attempt. For example, in figure 1, a Reflexion agent learns to optimize its own behavior to solve decision-making, programming, and reasoning tasks through trial, error, and self-reflection.

Generating useful reflective feedback is challenging since it requires a good understanding of where the model made mistakes (i.e. the credit assignment problem [25]) as well as the ability to generate a summary containing actionable insights for improvement. We explore three ways for doing this – simple binary environment feedback, pre-defined heuristics for common failure cases, and self-evaluation such as binary classification using LLMs (decision-making) or self-written unit tests (programming). In all implementations, the evaluation signal is amplified to natural language experience summaries which can be stored in long-term memory.

Reflexion has several advantages compared to more traditional RL approaches like policy or value-based learning: 1) it is lightweight and doesn't require finetuning the LLM, 2) it allows for more nuanced forms of feedback (e.g. targeted changes in actions), compared to scalar or vector rewards that are challenging to perform accurate credit assignment with, 3) it allows for a more explicit and interpretable form of episodic memory over prior experiences, and 4) it provides more explicit hints for actions in future episodes. At the same time, it does have the disadvantages of relying on the power of the LLM's self-evaluation capabilities (or heuristics) and not having a formal guarantee for success. However, as LLM capabilities improve, we only expect this paradigm to get better over time.

We perform experiments on (1) decision-making tasks to test sequential action choices over long trajectories, (2) reasoning tasks to test knowledge-intensive, single-step generation improvement, and (3) programming tasks to teach the agent to effectively use external tools such as compilers and interpreters. Across all three types of tasks, we observe Reflexion agents are better decision-makers, reasoners, and programmers. More concretely, Reflexion agents improve on decision-making AlfWorld [24] tasks over strong baseline approaches by an absolute 22% in 12 iterative learning steps, and on reasoning questions in HotPotQA [28] by 20%, and Python programming tasks on HumanEval [6] by as much as 11%.

To summarize, our contributions are the following:

- We propose Reflexion, a new paradigm for 'verbal' reinforcement that parameterizes a policy as an agent's memory encoding paired with a choice of LLM parameters.

- We explore this emergent property of *self-reflection* in LLMs and empirically show that self-reflection is extremely useful to learn complex tasks over a handful of trials.

- We introduce LeetcodeHardGym, a code-generation RL gym environment consisting of 40 challenging Leetcode questions ('hard-level') in 19 programming languages.

- We show that Reflexion achieves improvements over strong baselines across several tasks, and achieves state-of-the-art results on various code generation benchmarks.

| Related work on reasoning and decision-making | | | | | |
|---|---|---|---|---|---|
| **Approach** | **Self refine** | **Hidden constraints** | **Decision making** | **Binary reward** | **Memory** |
| Self-refine [15] | ✓ | ✗ | ✗ | ✗ | ✗ |
| Beam search [27] | ✓ | ✓ | ✓ | ✓ | ✗ |
| **Reflexion (ours)** | ✓ | ✓ | ✓ | ✓ | ✓ |
| Related work on programming | | | | | |
| **Approach Test execution** | **Test execution** | **Debugging** | **Self-generated tests** | **Multiple languages** | **Self-reflection** |
| AlphaCode [14] | ✓ | ✗ | ✗ | ✓ | ✗ |
| CodeT [5] | ✓ | ✗ | ✓ | ✗ | ✗ |
| Self-debugging [7] | ✓ | ✓ | ✗ | ✗ | ✗ |
| CodeRL [12] | ✓ | ✓ | ✗ | ✗ | ✗ |
| **Reflexion (ours)** | ✓ | ✓ | ✓ | ✓ | ✓ |

## 2 Related work

**Reasoning and decision-making** Self-Refine [15] employs an iterative framework for self-refinement to autonomously improve generation through self-evaluation. These self-evaluation and self-improvement steps are conditioned on given task constraints, such as "How can this generation be written in a more positive way". Self-Refine is effective but is limited to single-generation reasoning tasks. Pryzant et al. [21] performs a similar semantic prompt-writing optimization, but is also limited to single-generation tasks. Paul et al. [20] fine-tune critic models to provide intermediate feedback within trajectories to improve reasoning responses. Xie et al. [27] use stochastic beam search over actions to perform a more efficient decision-making search strategy which allows the agent to use foresight advantage due to its self-evaluation component. Yoran et al. [31] and Nair et al. [16] use decider models to reason over several generations. Kim et al. [10] use a retry pattern over a fixed number of steps without an evaluation step. Goodman [9] perform a qualitative evaluation step that proposes optimizations to the previous generation. In this paper, we show that several of these concepts can be enhanced with *self-reflection* to build a persisting memory of self-reflective experiences which allows an agent to identify its own errors and self-suggest lessons to learn from its mistakes over time.

**Programming** Several past and recent works employ variations of test-driven development or code debugging practices. AlphaCode [14] evaluates a set of generations on hidden test cases. CodeT [5] uses self-generated unit tests that are used to score generated function implementations. Self-Debugging [7] employs a debugging component that is used to improve existing implementations given feedback from a code execution environment. CodeRL [12] sets the problem in an RL framework using an actor-critic setup to debug programs given feedback from an execution environment. AlphaCode, Self-Debugging and CodeRL are effective in fixing less-complex program bugs, but they rely upon ground truth test cases that invalidate pass@1 eligibility, and do not use self-reflection to bridge the gap between error identification and implementation improvement. CodeT does not access hidden test cases but does not implement a self-learning step to improve code writing.

## 3 Reflexion: reinforcement via verbal reflection

We develop a modular formulation for Reflexion, utilizing three distinct models: an *Actor*, denoted as $M_a$, which generates text and actions; an *Evaluator* model, represented by $M_e$, that scores the outputs produced by $M_a$; and a *Self-Reflection* model, denoted as $M_{sr}$, which generates verbal reinforcement cues to assist the Actor in self-improvement. We provide a detailed description of each of these models and subsequently elucidate their collaborative functioning within the Reflexion framework.

**Actor** The Actor is built upon a large language model (LLM) that is specifically prompted to generate the necessary text and actions conditioned on the state observations. Analogous to traditional policy-based RL setups, we sample an action or generation, $a_t$, from the current policy $\pi_\theta$ at time $t$, receive an observation from the environment $o_t$. We explore various Actor models, including Chain of Thought [26] and ReAct [30]. These diverse generation models allow us to explore different aspects

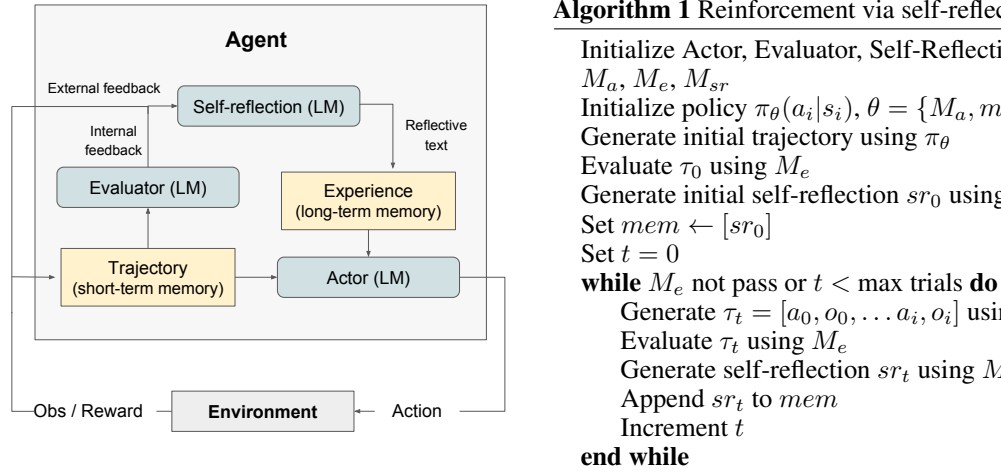

Figure 2: (a) Diagram of Reflexion. (b) Reflexion reinforcement algorithm

of text and action generation within the Reflexion framework, providing valuable insights into their performance and effectiveness. In addition, we also add a memory component *mem* that provides additional context to this agent. This adaption was inspired by Brooks et al. [3], who suggest a policy iteration approach using in-context learning. Details on how this is populated are provided below.

**Evaluator** The Evaluator component of the Reflexion framework plays a crucial role in assessing the quality of the generated outputs produced by the Actor. It takes as input a generated trajectory and computes a reward score that reflects its performance within the given task context. Defining effective value and reward functions that apply to semantic spaces is difficult, so we investigate several variants of the Evaluator model. For reasoning tasks, we explore reward functions based on exact match (EM) grading, ensuring that the generated output aligns closely with the expected solution. In decision-making tasks, we employ pre-defined heuristic functions that are tailored to specific evaluation criteria. Additionally, we experiment with using a different instantiation of an LLM itself as an Evaluator, generating rewards for decision-making and programming tasks. This multi-faceted approach to Evaluator design allows us to examine different strategies for scoring generated outputs, offering insights into their effectiveness and suitability across a range of tasks.

**Self-reflection** The Self-Reflection model instantiated as an LLM, plays a crucial role in the Reflexion framework by generating verbal self-reflections to provide valuable feedback for future trials. Given a sparse reward signal, such as a binary success status (success/fail), the current trajectory, and its persistent memory *mem*, the self-reflection model generates nuanced and specific feedback. This feedback, which is more informative than scalar rewards, is then stored in the agent's memory (*mem*). For instance, in a multi-step decision-making task, when the agent receives a failure signal, it can infer that a specific action $a_i$ led to subsequent incorrect actions $a_{i+1}$ and $a_{i+2}$. The agent can then verbally state that it should have taken a different action, $a'_i$, which would have resulted in $a'_{i+1}$ and $a'_{i+2}$, and store this experience in its memory. In subsequent trials, the agent can leverage its past experiences to adapt its decision-making approach at time $t$ by choosing action $a'_i$. This iterative process of trial, error, self-reflection, and persisting memory enables the agent to rapidly improve its decision-making ability in various environments by utilizing informative feedback signals.

**Memory** Core components of the Reflexion process are the notion of short-term and long-term memory. At inference time, the Actor conditions its decisions on short and long-term memory, similar to the way that humans remember fine-grain recent details while also recalling distilled important experiences from long-term memory. In the RL setup, the trajectory history serves as the short-term memory while outputs from the Self-Reflection model are stored in long-term memory. These two memory components work together to provide context that is specific but also influenced by lessons

learned over several trials, which is a key advantage of Reflexion agents over other LLM action choice works.

**The Reflexion process** Reflexion is formalized as an iterative optimization process in 1. In the first trial, the Actor produces a trajectory $\tau_0$ by interacting with the environment. The Evaluator then produces a score $r_0$ which is computed as $r_t = M_e(\tau_0)$. $r_t$ is only a scalar reward for trial $t$ that improves as task-specific performance increases. After the first trial, to amplify $r_0$ to a feedback form that can be used for improvement by an LLM, the Self-Reflection model analyzes the set of $\{\tau_0, r_0\}$ to produce a summary $sr_0$ which is stored in the memory *mem*. $sr_t$ is a verbal experience feedback for trial $t$. The Actor, Evaluator, and Self-Reflection models work together through trials in a loop until the Evaluator deems $\tau_t$ to be correct. As mentioned in 3, the memory component of Reflexion is crucial to its effectiveness. After each trial $t$, $sr_t$, is appended *mem*. In practice, we bound *mem* by a maximum number of stored experiences, $\Omega$ (usually set to 1-3) to adhere to max context LLM limitations.

# 4 Experiments

We evaluate various natural language RL setups on decision-making, reasoning, and code generation tasks. Specifically, we challenge an agent to perform search-based question answering on HotPotQA [28], multi-step tasks in common household environments in AlfWorld [24], and code writing tasks in competition-like environments with interpreters and compilers in HumanEval [6], MBPP [2], and LeetcodeHard, a new benchmark. Most notably, Reflexion improves performance over strong baselines by 22% in AlfWorld, 20% in HotPotQA, and 11% on HumanEval.

## 4.1 Sequential decision making: ALFWorld

AlfWorld is a suite of text-based environments that challenge an agent to solve multi-step tasks in a variety of interactive environments based on TextWorld [8]. Following Yao et al. [30], we run the agent in 134 AlfWorld environments across six different tasks, including finding hidden objects (e.g., finding a spatula in a drawer), moving objects (e.g., moving a knife to the cutting board), and manipulating objects with other objects (e.g., chilling a tomato in the fridge). We use ReAct [30] as the action generator as Yao et al. [30] has shown success in long trajectory decision-making using explicit intermediate thoughts. AlfWorld tasks naturally require a self-evaluation step as the environment can only signal if a task is complete. To achieve fully autonomous behavior, we implement two self-evaluation techniques: natural language classification using an LLM and a hand-written heuristic. The heuristic is simple: if the agent executes the same action and receives the same response for more than 3 cycles, or if the number of actions taken in the current environment exceeds 30 (inefficient planning), we self-reflect. In the baseline runs, if self-reflection is suggested, we skip the self-reflection process, reset the environment, and start a new trial. In the Reflexion runs, the agent uses self-reflection to find its mistake, update its memory, reset the environment, and start a new trial. To avoid very long prompt windows that may exceed the maximum limit, we truncate the agent's memory to the last 3 self-reflections (experiences).

To avoid syntactic errors, we provide two domain-specific few-shot trajectories to the agent. We use the same few-shot trajectory examples as Yao et al. [30] with GPT-3 for the LLM. AlfWorld tasks, ReAct few-shot prompts, and Reflexion examples are included in the appendix.

**Results** ReAct + Reflexion significantly outperforms ReAct by completing 130 out of 134 tasks using the simple heuristic to detect hallucinations and inefficient planning. Further, ReAct + Reflexion learns to solve additional tasks by learning in 12 consecutive trials. In the ReAct-only approach, we see that performance increase halts between trials 6 and 7.

**Analysis** A common error in baseline failed AlfWorld trajectories is when an agent thinks that it has possession of an item but does not actually have the item. The agent proceeds to execute several actions in a long trajectory and is not able to backtrack its actions to find the mistake. Reflexion eliminates almost all of these cases by using self-reflection to distill long, failed trajectories into relevant experiences that can are used as "self-hints" in the future. There are two main cases in which long-term memory helps an agent in AlfWorld: 1) An early mistake in a long trajectory can be easily identified. The agent can suggest a new action choice or even a new long-term plan. 2) There are too

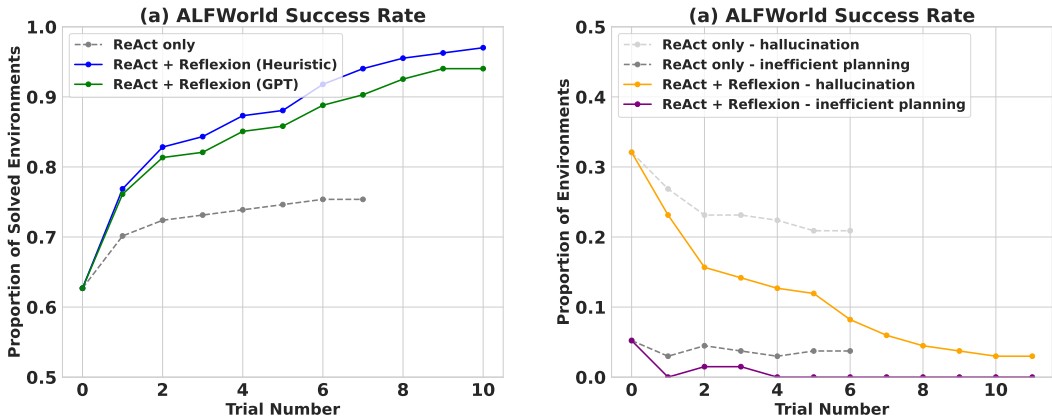

Figure 3: (a) AlfWorld performance across 134 tasks showing cumulative proportions of solved tasks using self-evaluation techniques of (Heuristic) and (GPT) for binary classification. (b) Classification of AlfWorld trajectories by reason of failure.

many surfaces/containers to check for an item. The agent can exploit its experience memory over several trials to thoroughly search a room. In 3, the learning curve suggests that the learning process occurs over several experiences, meaning that the agent is successfully balancing cases 1 and 2 shown in the immediate spike in the improvement between the first two trials, then a steady increase over the next 11 trials to a near-perfect performance. On the other hand, 3 shows a ReAct-only agent converging at a hallucination rate of 22% with no signs of long-term recovery.

## 4.2 Reasoning: HotpotQA

HotPotQA [28] is a Wikipedia-based dataset with 113k question-and-answer pairs that challenge agents to parse content and reason over several supporting documents. To test improvement in reasoning *only* ability, we implement Reflexion + Chain-of-Thought (CoT) [26] for step-by-step $Q \rightarrow A$ and $Q, C_{gt} \rightarrow A$ implementations, where $Q$ is the question, $C_{gt}$ is the ground truth context from the dataset, and $A$ is the final answer. Since CoT is not a multi-step decision-making technique, we give $C_{gt}$ to the agent so that we can isolate the reasoning behavior over large sections of the provided text. To test holistic question and answering ability, which requires reasoning and action choice, we implement a Reflexion + ReAct [30] agent that can retrieve relevant context using a Wikipedia API and infer answers using step-by-step explicit thinking. For CoT implementations, we use 6-shot prompting; for ReAct, we use 2-shot prompting, and for self-reflection, we use 2-shot prompting. All examples can be found in the appendix.

Robustly evaluating natural language answers is a long-standing problem in NLP. Therefore, between trials, we use exact match answer grading using the environment to give a binary success signal to the agent. After each trial, the self-reflection loop is employed to amplify the binary signal, similar to the decision-making setup 4.1 in AlfWorld with a memory size of 3 experiences.

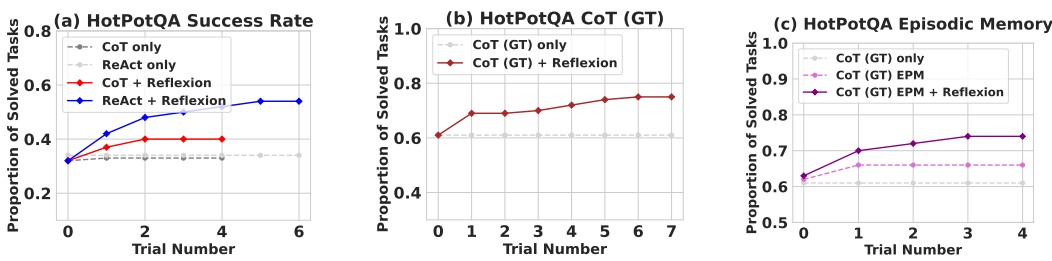

Figure 4: Chain-of-Thought (CoT) and ReAct. Reflexion improves search, information retrieval, and reasoning capabilities on 100 HotPotQA questions. (a) Reflexion ReAct vs Reflexion CoT (b) Reflexion CoT (GT) for reasoning only (c) Reflexion vs episodic memory ablation.

**Results**  Reflexion outperforms all baseline approaches by significant margins over several learning steps. Furthermore, ReAct-only, CoT-only, and CoT (GT)-only implementations fail to probabilistically improve on any tasks, meaning that no failed tasks from the first trial from any of the baseline approaches were able to be solved in subsequent trials using a temperature of 0.7 In the Reflexion runs, we allowed the agent to gather experience and retry on failed tasks until it produced 3 consecutive failed attempts on the particular task. Naturally, the CoT (GT) achieved higher accuracy scores as it was given access to the ground truth context of the question. Still, the CoT (GT) agent is unable to correctly infer the correct answer for 39% of the questions, but Reflexion helps the agent to correct its mistakes without access to the ground truth answer to improve its accuracy by 14%.

**Analysis**  We perform an ablation experiment to isolate the advantage of the self-reflective step for reasoning using CoT (GT) as the baseline approach 4. Recall that CoT (GT) uses Chain-of-Thought reasoning with provided ground truth context, which tests reasoning ability over long contexts. Next, we add an element of episodic memory (EPM) by including the most recent trajectory. For the Reflexion agent, we implement the standard self-reflection step as a final pass. Intuitively, we test if the agent is iteratively learning more effectively by using verbal explanation using language written in the first person. 4 shows that self-reflection improves learning by an 8% absolute boost over the episodic memory learning advantage. This result supports the argument that refinement-only approaches are not as effective as self-reflection-guided refinement approaches.

## 4.3  Programming

We evaluate the baseline and Reflexion approaches on Python and Rust code writing on MBPP [2], HumanEval [6], and LeetcodeHardGym, our new dataset. MBPP and HumanEval measure function body generation accuracy given natural language descriptions. We use a benchmark language compiler, MultiPL-E [4], to translate subsets of HumanEval and MBPP to the Rust language. MultiPL-E is a collection of small compilers that can be used to translate Python benchmark questions to 18 other languages. We include experiments for Rust code generation to demonstrate that Reflexion implementations for code generation are language-agnostic and can be used for interpreted and compiled languages. Lastly, we introduce a new benchmark, LeetcodeHardGym, which is an interactive programming gym that contains 40 Leetcode hard-rated questions that have been released after October 8, 2022, which is the pre-training cutoff date of GPT-4 [18].

The task of programming presents a unique opportunity to use more grounded self-evaluation practices such as self-generated unit test suites. Thus, our Reflexion-based programming task implementation is eligible for pass@1 accuracy reporting. To generate a test suite, we use Chain-of-Thought prompting [26] to produce diverse, extensive tests with corresponding natural language descriptions. Then, we filter for syntactically valid test statements by attempting to construct a valid abstract syntax tree (AST) for each proposed test. Finally, we sample $n$ tests from the collection of generated unit tests to produce a test suite $T$, denoted as $\{t_0, t_1, \ldots, t_n\}$. We set $n$ to a maximum of 6 unit tests. Aside from the unit test suite component, the setup for the learning loop for a Reflexion programming agent is identical to the reasoning and decision-making agents with a max memory limit of 1 experience.

| Benchmark + Language | Prev SOTA Pass@1 | SOTA Pass@1 | Reflexion Pass@1 |
|---|---|---|---|
| HumanEval (PY) | 65.8 (CodeT [5] + GPT-3.5) | 80.1 (GPT-4) | **91.0** |
| HumanEval (RS) | – | 60.0 (GPT-4) | **68.0** |
| MBPP (PY) | 67.7 (CodeT [5] + Codex [6]) | **80.1** (GPT-4) | 77.1 |
| MBPP (RS) | – | 70.9 (GPT-4) | **75.4** |
| Leetcode Hard (PY) | – | 7.5 (GPT-4) | **15.0** |

Table 1: Pass@1 accuracy for various model-strategy-language combinations. The base strategy is a single code generation sample. All instruction-based models follow zero-shot code generation.

**Results**  Reflexion outperforms all baseline accuracies and sets new state-of-the-art standards on all benchmarks for Python and Rust except for MBPP Python 1. We further investigate the inferior performance of Reflexion on MBPP Python.

**Analysis**  We acknowledge that self-reflecting code-generation agents are bound to their ability to write diverse, comprehensive tests. Therefore, in the case in which the model generates a flaky test

| Benchmark + Language | Base | Reflexion | TP | FN | FP | TN |
|---|---|---|---|---|---|---|
| HumanEval (PY) | 0.80 | **0.91** | 0.99 | 0.40 | 0.01 | 0.60 |
| MBPP (PY) | **0.80** | 0.77 | 0.84 | 0.59 | 0.16 | 0.41 |
| HumanEval (RS) | 0.60 | **0.68** | 0.87 | 0.37 | 0.13 | 0.63 |
| MBPP (RS) | 0.71 | **0.75** | 0.84 | 0.51 | 0.16 | 0.49 |

Table 2: Overall accuracy and test generation performance for HumanEval and MBPP. For Rust, HumanEval is the hardest 50 problems from HumanEval Python translated to Rust with MultiPL-E [4]. TP: unit tests pass, solution pass; FN: unit tests fail, solution pass; FP: unit tests pass, solution fail; TN: unit tests fail, solution fail.

suite, it is possible that all tests pass on an incorrect solution and lead to a false positive label on a code completion [11]. On the other hand, if the model produces an incorrectly written test suite, it is possible for some of the tests to fail on a correct solution, leading to a self-reflection generation that is conditioned on a false negative code completion. Given the implementation of Reflexion, false negatives are preferred over false positives as the agent may be able to use self-reflection to identify the incorrect test(s) and prompt itself to keep the original code completion intact. On the other hand, if an invalid test suite returns a false positive completion (all internal test cases pass but the implementation is incorrect), the agent will prematurely report an invalid submission. In 2, various conditions are measured to analyze performance beyond pass@1 accuracy. Previously, we displayed the inferior performance of Reflexion to the baseline GPT-4 on MBPP Python. In 2, we observe a notable discrepancy between the false positive labels produced by internal test execution, P(not pass@1 generation correct | tests pass). That is, the probability that a submission will fail given that it passes all unit tests. For HumanEval and MBPP Python, the baseline pass@1 accuracies are relatively similar, 82% and 80%, respectively. However, the false positive test execution rate for MBPP Python is 16.3% while the rate for HumanEval Python is a mere 1.4%, leading to 91% overall accuracy 1.

| Approach | Test Generation | Self-reflection | Pass@1 (Acc) |
|---|---|---|---|
| Base model | False | False | 0.60 |
| Test generation omission | False | True | 0.52 |
| Self-reflection omission | True | False | 0.60 |
| **Reflexion** | True | True | **0.68** |

Table 3: Pass@1 accuracy for various compromised approaches on the Reflexion approach using GPT-4 as the base model on HumanEval Rust - 50 hardest problems

**Ablation study**    We test the composite approach of Reflexion for test generation and self-reflection cooperation on a subset of the 50 hardest HumanEval Rust problems. Our Rust compiler environment provides verbose error logs and helpful debugging hints, therefore serving as a good playground for compromised approaches. First, we omit internal test generation and execution steps, which test the agent to self-reflect without guidance from current implementations. 3 shows an inferior 52% vs 60% (baseline) accuracy, which suggests that the agent is unable to determine if the current implementation is correct without unit tests. Therefore, the agent must participate in all iterations of the run without the option to return early, performing harmful edits to the implementation.

Next, we test self-reflection contribution by omitting the natural language explanation step following failed unit test suite evaluations. Intuitively, this challenges the agent to combine the tasks of error identification and implementation improvement across all failed unit tests. Interestingly, the compromised agent does not improve performance over the baseline run. We observe that the test generation and code compilation steps are able to catch syntax and logic errors, but the implementation fixes do not reflect these indications. These empirical results suggest that several recent works that propose *blind* trial and error debugging techniques without self-reflection are ineffective on harder tasks such as writing complex programs in Rust.

## 5   Limitations

At its core, Reflexion is an optimization technique that uses natural language to do policy optimization. Policy optimization is a powerful approach to improve action choice through experience, but it may still succumb to non-optimal local minima solutions. In this study, we limit long-term memory to a sliding window with maximum capacity, but we encourage future work to extend the memory component of *Reflexion* with more advanced structures such as vector embedding databases or traditional SQL databases. Specific to code generation, there are many practical limitations to test-driven development in specifying accurate input-output mappings such as non-deterministic generator functions, impure functions that interact with APIs, functions that vary output according to hardware specifications, or functions that invoke parallel or concurrent behavior that may be difficult to predict.

## 6   Broader impact

Large language models are increasingly used to interact with external environments (e.g. the Internet, software, robotics, etc.) and humans. Our work has the potential of reinforcing and empowering these agents toward greater automation and work efficiency, but it also amplifies the risks when these agents were put into misuse. We believe that this direction of research will need more effort in safety and ethical considerations.

On the other hand, reinforcement learning has suffered from its black-box policy and optimization setups in which interpretability and alignment have been challenging. Our proposed "verbal" reinforcement learning might address some of the issues and turn autonomous agents more interpretable and diagnosable. For example, in the case of tool-usage that may be too hard for humans to understand, self-reflections could be monitored to ensure proper intent before using the tool.

## 7   Conclusion

In this work, we present *Reflexion*, an approach that leverages verbal reinforcement to teach agents to learn from past mistakes. We empirically show that Reflexion agents significantly outperform currently widely-used decision-making approaches by utilizing self-reflection. In future work, Reflexion could be used to employ more advanced techniques that have been thoroughly studied in traditional RL settings, such as value learning in natural language or off-policy exploration techniques.

## 8   Reproducibility

We highly advise others to use isolated execution environments when running autonomous code writing experiments as the generated code is not validated before execution.

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

# A  Evaluation with additional models

We further investigated the applicability of trial-and-error problem-solving with models of various strengths. We found that the ability to specify self-corrections is an emergent quality of stronger, larger models.

| Approach | Pass@1 accuracy (avg over 8 trials) | Pass@1 accuracy (std) |
|---|---|---|
| Baseline | 0.26 | 0.00481 |
| Reflexion | 0.26 | 0.00305 |

Table 4: Pass@1 accuracy on HumanEval Python using starchat-beta [13].

| Model | Baseline accuracy | Reflexion accuracy |
|---|---|---|
| CoT (GT) + text-davinci-003 | 0.60 | **0.77** |
| CoT (GT) + gpt-3.5-turbo | 0.57 | **0.71** |
| CoT (GT) + gpt-4 | 0.68 | **0.80** |
| ReAct + text-davinci-003 | 0.30 | **0.55** |
| ReAct + gpt-3.5-turbo | 0.26 | **0.38** |
| ReAct + gpt-4 | 0.39 | **0.51** |

Table 5: Pass@1 accuracy on 100 HotPotQA using various models.

# B   Decision-making

**Environment:** You are in the middle of a room. Looking quickly around you, you see a bed 1, a desk 2, a desk 1, a drawer 6, a drawer 5, a drawer 4, a drawer 3, a drawer 2, a drawer 1, a garbagecan 1, a laundryhamper 1, a safe 1, a shelf 6, a shelf 5, a shelf 4, a shelf 3, a shelf 2, and a shelf 1.
**Task:** examine the mug with the desklamp.
**Trial #1**
> think: To solve the task, I need to find and take a mug, then find and use a desklamp.
> think: First I need to find a mug. A mug is more likely to appear in drawer (1-6), desk (1-2), shelf (1-6), garbagecan (1), laundryhamper (1). I can check one by one, starting with drawer 1.
> go to drawer 1
The drawer 1 is closed.
. . .
> open drawer 6
You open the drawer 6. The drawer 6 is open. In it, you see a keychain 2.
> go to desk 1
On the desk 1, you see a creditcard 3, a desklamp 1, a laptop 2, a mug 1, a pen 1, and a pencil 1.
> think: Now I find a mug (1). Next, I need to take it.
> take mug 1 from desk 1
You pick up the mug 1 from the desk 1.
> think: Now I take a mug (1). Next, I need to find a desklamp. A desklamp is more likely to appear in desk (1-2), sidetable (1-2), shelf (1-6), bed (1), drawer (1-6). I can check one by one, starting with desk 1.
> go to desk 1
Nothing happens.
> go to desk 2
On the desk 2, you see a alarmclock 1, a bowl 1, a mug 3, a pencil 3, and a pencil 2.
> think: Now I find a desklamp (1). Next, I need to use it.
> use desklamp 1
Nothing happens.
> use desklamp 1
Nothing happens.
Status: Fail
**Reflection:** In this environment, my plan was to find a mug then find and use a desklamp. However, the task says to examine the mug with the desklamp. I should have looked for the desklamp first, then looked for the mug. I noticed that the desklamp was found on desk 1. In the next trial, I will go to desk 1, find the lamp, then look for the mug and examine it with the desklamp.
**Trial #2**
> go to desk 1
On the desk 1, you see a creditcard 3, a desklamp 1, a laptop 2, a mug 1, a pen 1, and a pencil 1.
> think: To solve the task, I need to find and take a mug, then find and use a desklamp.
> take mug 1 from desk 1
You pick up the mug 1 from the desk 1.
> think: To solve the task, I need to find and take a mug, then find and use a desklamp.
> use desklamp 1
You turn on the desklamp 1.
Status: Success

Figure 5: [Top] An AlfWorld trajectory in which the agent failed due to inefficient planning. In the reflection, the agent recognizes that it should have looked for the desklamp then the mug, not the mug then the desklamp. [Bottom] The agent is able to correct its reasoning trace and execute a sequence of actions in a concise manner.

### B.1 WebShop Limitation

In 5, we briefly state that Reflexion struggles to overcome local minima choices that require extremely creative behavior to escape. We observe this shortcoming in an experiment on WebShop [29]. WebShop is a web-based problem-solving benchmark that tests agents to navigate an e-commerce website to locate and purchase products given requests from clients. We test a two-shot ReAct + Reflexion agent in 100 environments. However, after only four trials, we terminate the runs as the agent does not show signs of improvement 6. Further, the agent does not generate helpful, intuitive self-reflections after failed attempts. We conclude that Reflexion is unable to solve tasks that require a significant amount of diversity and exploration. In AlfWorld, the agent is able to adequately explore new environments because the permissible actions can be seen in the observations. In HotPotQA, the agent faces a similar WebShop search query task but is more successful as the search space for Wikipedia articles is more diverse and requires less precise search queries. A common problem for e-commerce search engines is properly handling ambiguity in natural language search interpretations. Thus, WebShop presents a task that requires very diverse and unique behavior from a Reflexion agent.

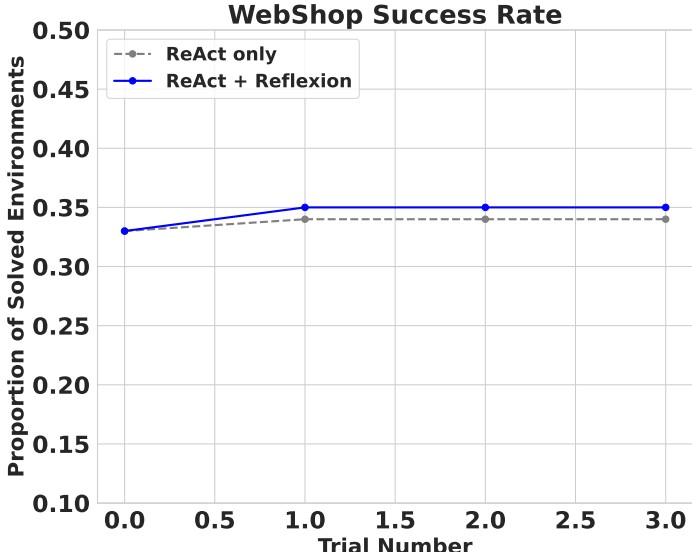

Figure 6: Reflexion vs React performance on WebShop across 100 customer shopping requests. ReAct + Reflexion fails to significantly outperform ReAct.

## C Programming

Programming LLM calls require strict instructions to produce function bodies *only*, due to the extensive dialogue training of the LLMs. A few programming examples are reported below with instructions highlighted in blue and templates. See the full implementation at `https://github.com/noahshinn024/reflexion`.

### C.1 Programming function implementation example (HumanEval Python)

Sample function signature:

```python
def minSubArraySum(nums):
    """
    Given an array of integers nums, find the minimum sum of
        any
    non-empty sub-array of nums.
    Example
    minSubArraySum([2, 3, 4, 1, 2, 4]) == 1
```

```
7       minSubArraySum([-1, -2, -3]) == -6
8       """
```

Sample function body implementation:

```python
1    min_sum = float('inf')
2    for i in range(len(nums)):
3        current_sum = 0
4        for j in range(i, len(nums)):
5            current_sum += nums[j]
6            if current_sum < min_sum:
7                min_sum = current_sum
8    return min_sum
```

## C.2   Reflexion Actor instruction

You are a Python writing assistant. You will be given your previous implementation of a function, a series of unit tests results, and your self-reflection on your previous implementation. Apply the necessary changes below by responding only with the improved body of the function. Do not include the signature in your response. The first line of your response should have 4 spaces of indentation so that it fits syntactically with the user provided signature. You will be given a few examples by the user.

Reflexion Actor generations follow the form:
(Instruction)
(Function implementation)
(Unit test feedback)
(Self-reflection)
(Instruction for next function implmentation)

Full examples are shown in `https://github.com/noahshinn024/reflexion`

## C.3   Reflexion Self-reflection instruction and example

You are a Python writing assistant. You will be given your previous implementation of a function, a series of unit tests results, and your self-reflection on your previous implementation. Apply the necessary changes below by responding only with the improved body of the function. Do not include the signature in your response. The first line of your response should have 4 spaces of indentation so that it fits syntactically with the user provided signature. You will be given a few examples by the user. Reflexion Self-Reflection generations follow the form:
(Instruction)
(Function implementation)
(Unit test feedback)

## C.4   Reflexion programming *no* Self-Reflection ablation example

Reflexion *no* Self-Reflection ablation Actor generations follow the form:
(Instruction)
(Function implementation)
(Unit test feedback)
~~(Self-reflection)~~
(Instruction for next function implmentation)

## C.5   Reflexion programming *no* test generation ablation example

Reflexion *no* test generation ablation Actor generations follow the form:
(Instruction)

(Function implementation)
~~(Unit test feedback)~~
(Self-reflection)
(Instruction for next function implmentation)

# D   Reasoning

## D.1   Full example

**Trial #1:** Question: Grown-Ups starred the actor who was best known for which role on "'Allo 'Allo!"?

Thought 1: I need to search Grown-Ups and "'Allo 'Allo!", find the actor who starred in Grown-Ups, then find the role he was best known for in "'Allo 'Allo!".

Action 1: Search[Grown-Ups]

Observation 1: Grown-Ups is a 1980 British BBC television film devised and directed by Mike Leigh. It stars Lesley Manville, Philip Davis, Brenda Blethyn, Janine Duvitski, Lindsay Duncan and Sam Kelly. It was edited by Robin Sales and produced by Louis Marks for the BBC, and originally shown on BBC 2 on 28 November 1980.

Thought 2: Grown-Ups starred Lesley Manville, Philip Davis, Brenda Blethyn, Janine Duvitski, Lindsay Duncan and Sam Kelly. I need to search "'Allo 'Allo!" and find which actor was best known for which role.

Action 2: Search["'Allo 'Allo!"]

Observation 2: Could not find ["'Allo 'Allo!"]. Similar: ["'Allo 'Allo!", "List of 'Allo 'Allo! characters", 'Gorden Kaye', "'Allo 'Allo! (series 2)", 'Allô, allô', 'Francesca Gonshaw', "'Allo 'Allo! (series 4)", 'Kim Hartman', "'Allo 'Allo! (series 6)", 'Jeremy Lloyd']

Thought 3: To find the actor who was best known for which role on "'Allo 'Allo!", I can search Gorden Kaye.

Action 3: Search[Gorden Kaye]

Observation 3: Gordon Irving Kaye (7 April 1941 – 23 January 2017), known professionally as Gorden Kaye, was an English actor, best known for playing womanising café owner René Artois in the television comedy series 'Allo 'Allo!.

Thought 4: Gorden Kaye was best known for playing womanising café owner René Artois in the television comedy series "'Allo 'Allo!". So the answer is Rene Artois.

Action 4: Finish[Rene Artois]

Observation 4: Answer is INCORRECT

**Reflection:** I searched the wrong title for the show, "'Allo 'Allo!", which resulted in no results. I should have searched the show's main character, Gorden Kaye, to find the role he was best known for in the show.

**Trial #2:** Question: Grown-Ups starred the actor who was best known for which role on "'Allo 'Allo!"?

Thought 1: I need to search the actor who was best known for a role on "'Allo 'Allo!" and find out what role they were best known for.

Action 1: Search[Grown-Ups]

Observation 1: Grown-Ups is a 1980 British BBC television film devised and directed by Mike Leigh. It stars Lesley Manville, Philip Davis, Brenda Blethyn, Janine Duvitski, Lindsay Duncan and Sam Kelly. It was edited by Robin Sales and produced by Louis Marks for the BBC, and originally shown on BBC 2 on 28 November 1980.

Thought 2: The paragraph does not mention the actor who was best known for a role on "'Allo 'Allo!". I need to search the actor's name instead.

Action 2: Search[Sam Kelly]

Observation 2: Roger Michael Kelly (19 December 1943 – 14 June 2014), known by the stage name Sam Kelly, was an English actor who appeared in film, television, radio and theatre. He is best known for his roles as Captain Hans Geering in 'Allo 'Allo!, Warren in Porridge, Sam in On the Up, and Ted Liversidge in Barbara.

Thought 3: Sam Kelly is best known for his role as Captain Hans Geering in "'Allo 'Allo!", so the answer is Captain Hans Geering.

Action 3: Finish[Captain Hans Geering]

Observation 3: Answer is CORRECT

Figure 7: Two HotPotQA trials within the same environment and task. The Reflexion + ReAct agent uses self-reflection to determine a better search method for the next trial.

## D.2 Chain-of-Thought + Reflexion

**Trial #1:** Question: What profession does John Lanchester and Alan Dean Foster have in common?
Thought 1: Let's think step by step. John Lanchester is a novelist, journalist, and critic. Alan Dean Foster is a novelist and screenwriter. So the profession John Lanchester and Alan Dean Foster have in common is novelist and screenwriter.
Action 1: Finish[novelist and screenwriter]
Observation 1: Answer is INCORRECT

**Trial #2:** Question: What profession does John Lanchester and Alan Dean Foster have in common?
Thought 1: Let's think step by step. John Lanchester is a novelist, journalist, and financial journalist. Alan Dean Foster is a novelist and screenwriter. So the profession John Lanchester and Alan Dean Foster have in common is novelist.
Action 1: Finish[novelist]
Observation 1: Answer is CORRECT

**Reflection:** My reasoning for the profession John Lanchester and Alan Dean Foster have in common failed because I incorrectly assumed that they both had the same profession. In the future, when attempting this question, I should focus on better researching the individual backgrounds of both authors to ensure that I am accurately identifying their professions. Additionally, I should consider the possibility that they may have multiple professions in common.

## D.3 HotPotQA Chain-of-Thought (GT) + Reflexion

**Context:** The Battle of White Plains was a battle in the New York and New Jersey campaign of the American Revolutionary War fought on October 28, 1776, near White Plains, New York. Following the retreat ...

**Trial #1:** Question: What was a series of battles during the Revolutionary War, for control of New York City and the state of New Jersey, fought on October 28, 1776 near White Plains, New York?
Thought 1: The question is asking about a series of battles during the Revolutionary War for control of New York City and the state of New Jersey, fought on October 28, 1776 near White Plains, New York. From the context, I know that this was the Battle of White Plains.
Action 1: Finish[Battle of White Plains]
Observation 1: Answer is INCORRECT

**Trial #2:** Question: What was a series of battles during the Revolutionary War, for control of New York City and the state of New Jersey, fought on October 28, 1776 near White Plains, New York?
Thought 1: The question is asking for a series of battles during the Revolutionary War, for control of New York City and the state of New Jersey, fought on October 28, 1776 near White Plains, New York. Based on the context, I know that this series of battles was the New York and New Jersey campaign.
Action 1: Finish[The New York and New Jersey campaign]
Observation 1: Answer is CORRECT

**Reflection:** Upon reflecting on the incorrect answer I provided, I realize that I may not have provided enough context to accurately answer the question. The question asked for a series of battles, but I only provided the name of one battle. To better answer this question in the future, I will make sure to provide more context, such as the name of the campaign, which was the New York and New Jersey campaign, and the fact that it was a series of battles. Additionally, I will make sure to include the dates of the battles, as well as the locations, to ensure that my answer is comprehensive and accurate.

## D.4 HotPotQA episodic memory (EPM) ablation prompts

### D.4.1 (EPM) Chain-of-Thought + Reflexion

**Trial #1:** Question: Which of Jonny Craig and Pete Doherty has been a member of more bands ?

Thought 1: Let's think step by step. Jonny Craig has been a member of six bands, including Emarosa, Dance Gavin Dance, and Slaves. Pete Doherty has been a member of seven bands, including The Libertines, Babyshambles, and The Puta Madres. So Pete Doherty has been a member of more bands.

Action 1: Finish[Pete Doherty]

Observation 1: Answer is INCORRECT

**Trial #2:** Question: Which of Jonny Craig and Pete Doherty has been a member of more bands ?

Thought 1: Let's think step by step. I will research the past and current bands of both musicians to ensure that I am accurately comparing their band memberships. After researching, I have found that Jonny Craig has been a member of seven bands, including Emarosa, Dance Gavin Dance, and Slaves. Pete Doherty has been a member of seven bands, including The Libertines, Babyshambles, and The Puta Madres. So Jonny Craig has been a member of more bands.

Action 2: Finish[Jonny Craig]

Observation 2: Answer is CORRECT

**Reflection:** My reasoning for which musician has been a member of more bands failed because I did not take into account that Jonny Craig has been a member of more bands in the past, even though Pete Doherty is currently a member of more bands. In the future, when attempting this question, I should focus on researching the past and current bands of both musicians to ensure that I am accurately comparing their band memberships.

### D.4.2 (EPM) Chain-of-Thought (GT) + Reflexion

**Context:** Hari Bahadur Basnet is a Nepalese politician. He is the head of the Foreign Relations Department of the Rastriya Janashakti Party. Basnet holds a M.Sc. in Engineering.
A Master of Science (Latin: "Magister Scientiae" ; abbreviated MS, M.S., MSc, M.Sc., MSci, M.Sci., ScM, Sc.M., SciM or Sci.M.) is a master's degree in the field of science...

**Trial #1:** Question: The head of the Foreign Relations Department of the Rastriya Janashakti Party holds a degree that can be abbreviated MS, M.S., or ScM, in what field?

(Previous trajectory)
Thought 1: The question is asking what field of study the head of the Foreign Relations Department of the Rastriya Janashakti Party holds a degree in. From the context, I know that the degree is a Master of Science degree, which is typically granted for studies in sciences, engineering, and medicine.

Action 1: Finish[Sciences, Engineering, and Medicine]

Observation 1: Answer is INCORRECT

**Trial #2:** Question: The head of the Foreign Relations Department of the Rastriya Janashakti Party holds a degree that can be abbreviated MS, M.S., or ScM, in what field?

(Previous trajectory)
Thought 1: The question is asking what field of study the head of the Foreign Relations Department of the Rastriya Janashakti Party holds a degree in. From the context, I know that the degree is a Master of Science degree, which is typically granted for studies in sciences, engineering, and medicine. However, the question is asking for the field of study, not the degree. Based on the context, I know that Hari Bahadur Basnet holds a M.Sc. in Engineering.

Action 1: Finish[Engineering]

Observation 1: Answer is CORRECT

**Reflection:** For the first trial, I misunderstood the question. I thought the question was asking for a category of degrees, rather than a specific degree. I learned from my mistake and in the second trial, I focused more closely on the question to understand that it was asking for a specific field of study.

