# OpenReview forum: "Reflexion: language agents with verbal reinforcement learning"
_NeurIPS.cc/2023/Conference — NeurIPS 2023 poster_

### Official Review · Reviewer_6XLB · 2023-06-23

**Soundness:** 4 excellent
**Presentation:** 4 excellent
**Contribution:** 3 good
**Rating:** 7
**Confidence:** 4

**Summary:**

This paper presents a method by which LLMs can improve task performance through a form of learning in language: distilling unsuccessful attempts down to a "lesson learned" via a self-reflection process. The work shows that this idea leads to improved performance within a variety of task domains, and includes fairly thorough ablations, which I appreciated.



**Strengths:**

I find this to be a fairly satisfying paper overall.
* The idea is compelling, and well executed.
* The experiments demonstrate that the approach is beneficial across three fairly diverse LM application domains, which shows that it is not simply a narrow improvement.
* The ablations are fairly thorough, and make the results more compelling.
* The paper is generally well-written and presented.

**Weaknesses:**

I have a few suggestions on how the paper could be improved:

1) I think this work could be more thoroughly integrated with the existing literature on learning from language feedback, both within LLMs (e.g. https://arxiv.org/abs/2204.14146 and https://arxiv.org/abs/2303.16749) and RL agents (https://proceedings.neurips.cc/paper_files/paper/2022/hash/0113ef4642264adc2e6924a3cbbdf532-Abstract-Conference.html and https://arxiv.org/abs/2112.03753). Another recent paper https://arxiv.org/abs/2304.03442 proposed a similar reflection approach, though with different motivation; it came out quite recently though so there is no expectation to refer to it.

2) Evaluations on GPT-4 are challenging to interpret. We don't know what data the model was trained or tuned on, and even if the datasets used were generated after the pre-training cutoff, the model may have encountered them in the fine-tuning or RLHF stages. Indeed, OpenAI explicitly uses information submitted to their chat (at least) as training data; can we guarantee that no LeetCoder submitted the problems to GPT-4 to check their solutions? This problem is exacerbated if the benchmark has previously been evaluated against GPT-4, as it seems these problems have; there have been substantial concerns about GPT-4's coding performance evaluations being affected by data contamination. It would be best if the authors at least acknowledge these issues in the manuscript; ideally, they would also evaluate against an open language model for which the training process is more transparent, although undoubtedly such a model might not perform as well.


**Questions:**

See above for some suggestions.

**Limitations:**

Limitations are discussed

---

> ### Author Rebuttal · Authors · 2023-08-10
>
>
> Thank you for your helpful suggestions!
>
> ### 1. Related work
>
> Thank you for pointing out these, which we will cite and discuss!
>
> ### 2. Evaluation for closed-source large language models
>
> - This is a general issue for all works that use closed-source LLMs, and motivated us to create our own LeetcodeHardGym Benchmark, which was specifically filtered to only contain problems created later than the training data cutoff date of GPT-4.
> - Across all tasks, baseline methods based on GPT-4 are far from perfect, and Reflexion + GPT-4 significantly improves the performance, which indicates GPT-4 might not have memorized these tasks, and better prompting methods might be important.
> - It is also possible to use open-source LLMs for Reflexion, e.g. see **General Response (2)**.

---

> > ### Comment · Reviewer_6XLB · 2023-08-10
> > **Thanks for the response**
> >
> > Thanks to the authors for the response!
> >
> > To clarify my comment on contamination: while I acknowledge that the new benchmark was created after the pretraining cutoff, we know that GPT-4's training has continued up to the present (at the very least to address vulnerabilities and introduce new features); it's quite reasonable to presume they use user chats for that training, and so it is still possible the model has been trained on these problems. The fact that it does better with Reflexion than alone is promising in that regard, but does not completely rule out the possibility (just because an LM has been trained on something does not mean it will always get it right). I still think it might be worth acknowledging this more explicitly, even if it is a general problem.
> >
> > However, I do want to reiterate that I think this is a valuable work that should certainly be accepted.

---

> > > ### Author Response · Authors · 2023-08-10
> > > **Agree**
> > >
> > > Yes, we agree this is a valid concern. We will explicitly acknowledge such an issue and our attempts toward tackling the issue (e.g. new leetcode benchmark) in the revision.
> > >
> > > Thanks again for finding our work valuable!

---

### Official Review · Reviewer_U48e · 2023-07-05

**Soundness:** 3 good
**Presentation:** 3 good
**Contribution:** 3 good
**Rating:** 6
**Confidence:** 4

**Summary:**

The paper introduces a novel framework, Reflexion, to improve the learning capabilities of language agents. Instead of traditional reinforcement learning methods, Reflexion uses linguistic feedback, where agents verbally reflect on task feedback and store these reflections in an episodic memory buffer to enhance decision-making. The authors demonstrate that Reflexion, which can incorporate various types and sources of feedback, outperforms baseline agents across diverse tasks, including text games, natural language question-answering, and code generation.

**Strengths:**

1. The paper introduces a unique framework, Reflexion, that uses linguistic feedback for reinforcement learning in language agents. This is a significant departure from traditional methods that rely on weight updates.
2. The paper offers a good explanation of their proposed methods and is easy to follow. Good visualizations are also included to help with reader understanding.
3. Abundant experiments have been conducted across 3 different tasks, covering text games, question answering, and code generation. A comprehensive analysis has also shown the effectiveness of the reflexion method.

**Weaknesses:**

1. Although reflexion shows significant improvements over baselines on three different tasks, I still feel that reflexion shows limited generalization ability on different tasks. The key modules in the framework, including evaluator and self-reflection, have varying designs for different tasks. In other words, the specific design details may only work on one task and may fail on the other ones.
2. The results for comparison in the question-answering task seem to be unfair. The reflexion framework leverages the environment binary reward of whether the answer is correct as the ground-truth answer or not. This leaks a little information from testing question-answer pairs. At least the ReAct baseline did not include such information.
3. There are some related works mentioned in Section 2. However, there is no comparison between Reflexion and these methods in the experiment sections.

**Questions:**

1. Can you offer some statistics on the running time of the reflexion framework? As it seems that the optimization process is time-consuming.
2. I am very curious about the comparison between reflexion and methods like self-refine and self-debugging.
3. Can you offer some detailed analysis of the differences between your method (reflexion) and simple ReAct+self-refine?
4. Is the quality of test cases generated in code generation tasks important? Are they guaranteed to be correct?


**Limitations:**

The authors have adequately introduced the limitations of their work.

---

> ### Author Rebuttal · Authors · 2023-08-10
>
>
> ### 1. the specific design details may only work on one task and may fail on the other ones
>
> Please see the General Response.
>
> ### 2. Accuracy comparisons
>
> You are right. The Reflexion performance on HotpotQA is not meant as a SoTA claim. Rather, it is intended to show how performances can be improved with multiple rounds of reflections. ALFWorld and Coding results are fair comparisons. We will better clarify it in the paper.
>
> ### 3. Comparison to other methods
> - Self-refine and self-debugging: self-refine is designed for text generation tasks, and self-debugging is designed for coding tasks. In contrast, Reflexion is designed for general agent tasks, which covers text generation, coding, and also embodied tasks (e.g. AlfWorld).
> - ReAct + self-refine: self-refine was not designed for acting tasks, cannot generated self-tests, and cannot incorporate environment or self-generated reward signals as Reflexion.
> - Experiments: In our programming ablation experiments, we omit various components, and these ablations are similar to baseline methods such as CodeT and Self-Debugging. Let us know if you think some other important baseline method is missing.
>
> ### 4. Running time statistics
>
> We evaluated `starchat-beta` on HumanEval Python - 161 problems (as mentioned in the General Response) and recorded wall time on 1 NVIDIA H100 GPU.
>
> *avg over 8 runs
> Simple: 00:25:18 (hh:mm:ss)
> Reflexion: 01:17:06 (hh:mm:ss)
>
> ### 5. Importance of tests
>
> Yes, the quality of the tests in code generation is important and a bottleneck. In **Table 2**, we show that the high false positive rate of MBPP Python tests (relative to HumanEval Python) largely contributed to the inferior performance of Reflexion.

---

> > ### Comment · Reviewer_U48e · 2023-08-18
> > **Thanks for the Response**
> >
> > I think most of my concerns have been addressed by the authors. Thanks for the efforts during the rebuttal. Although some of the points I listed in the review seem to be framework limitations, I still find the paper interesting and useful. I will maintain my positive score.

---

### Official Review · Reviewer_BKdQ · 2023-07-05

**Soundness:** 2 fair
**Presentation:** 3 good
**Contribution:** 3 good
**Rating:** 4
**Confidence:** 3

**Summary:**

This paper proposes a novel framework to do RL style learning through LLM. The idea is interesting. The authors have also done evaluation to show this technique/framework can work better than baseline methods w/ a few different tasks.

**Strengths:**

The author proposed a novel framework to let LLM learn new tasks through trial and error. The framework seems general and can handle multiple tasks.

**Weaknesses:**

1. While the framework is general. The devil is in the details. It seems for each specific task, it requires different heuristic and tuning. Also different task may require different LLM/VLM as actor/evaluator/self-reflection.

2. In terms of ablation, since this paper focuses on the framework, it seems more useful to ablate different framework level choices. i.e. is the self-reflection module really the key component in this framework, or can we bypass it by some other way, such as prompting?

**Questions:**

Will this framework also work well w/ VLMs? (if we also have vision/images as input)

**Limitations:**

See weakness section.

---

> ### Author Rebuttal · Authors · 2023-08-10
>
>
> ### 1. different heuristic/tuning/LLM/VLM for each task
>
> - In this work, there is no finetuning or VLM used.
> - We only used OpenAI GPT models as LLMs in the paper. In **General Response (2)**, we show it's also possible to use other LLMs, or use different LLMs for different roles (actor, evaluator, self-reflection). We believe such a flexibility is a benefit, rather than an issue.
> - Prompt design is task-specific to some degree, which is true for all prompting methods. However, Reflexion allows a principled way to design prompted roles (actor, evaluator, self-reflection) and their control flow. Also, the self-reflection prompt is mostly similar across different tasks.
> - In summary, as you said, "the framework is general" to solve diverse tasks because it could flexibly incorporate task heuristics with principled prompt role design and control flow.
>
> ### 2. Necessity for self-reflection
>
> - We cannot "bypass self-reflection by prompting", because self-reflection is done via prompting in our work.
> - Ablating self-reflection is equivalent to ReAct/CoT baselines, which we reported in all tasks.
>
> ### 3. Possible application with VLMs
>
> We believe the idea of generating and conditioning on text reflections should be possible and useful for any language-conditioned policies, including ones using VLMs. This is an interesting question and a good future direction.

---

### Official Review · Reviewer_V8mX · 2023-07-06

**Soundness:** 3 good
**Presentation:** 3 good
**Contribution:** 4 excellent
**Rating:** 7
**Confidence:** 4

**Summary:**

The paper presents Reflexion, a framework to reinforce language agents by using language feedback. Reflexion agents verbally reflect on task feedback signals, then store their own reflective text in a memory buffer to improve their decision-making in future trials. Reflexion is flexible and effective across diverse tasks (sequential decision-making, coding, language reasoning). The paper also explores different types and sources of feedback signals, feedback incorporation methods, and agent types, and provides insights into how they affect performance.

**Strengths:**

1. This paper proposes Reflexion, a new framework for “verbal” reinforcement that conditions a policy on the agent’s “memory” which is beneficial for the development of LLMs, reinforcement learning, reasoning, code generation, and so on.
2. The results across diverse tasks including sequential decision-making, language reasoning, and programming are very promising with a large advantage.
3. This paper proposes a new benchmark, LeetcodeHardGym, consisting of 40 challenging Leetcode questions in 19 different programming languages, which is beneficial for the program synthesis and code generation community.

**Weaknesses:**

1. Various LLMs (the GPT family, the vicuna family, and models with different sizes) should be evaluated to show reflexion’s generalization ability across models. Although reflexion performs well, it is important to analyze its border.
2. Ablation study about long/short term memory. For example, in ALFWorld the baseline is react which resets the environment and starts a new trial whenever self-reflexion is suggested. I take it as “reflexion w/o long/short term memory”. Then, how about the performance of “reflexion w/o long-term memory (but w/ short-term memory)” and the performance of “reflexion w/o short-term memory (but w/ long-term memory)”?

**Questions:**

1. According to line 117, the evaluator is not necessarily an LLM, right? But it is pictured as “LM” in figure 2 which is a little bit confusing.
2. In algorithm 1, $\pi_\theta$ is not relevant to trajectory $\tau_{t-1}$, the short-term memory. But in figure 2 (a) and section 3, the Actor receives short-term memory as its input.
3. How does short-term memory work on programming tasks? Is reflexion = self-debugging + CodeT?

**Limitations:**

The author has discussed the limitations well.

---

> ### Author Rebuttal · Authors · 2023-08-10
>
> Thank you for the great comments!
>
> ### 1. Other models
>
> Please see **General Response (2)**, where we run additional experiments on closed and open source LLMs.
>
> ### 2. Short vs long-term memory
>
> In this work, we define short-term memory as the current trajectory and long-term memory as the reflections. It is not possible to omit short-term memory, and a long term memory ablation would be the baseline ReAct/CoT experiment.
>
> ### 3. Evaluator
>
> Yes, the Evaluator does not have to be an LLM. We will revise the figure.
>
> ### 4. Policy
>
> Yes, the policy is conditioned on the trajectory (and possibly other reflections). We will update the algorithm specification.
>
> ### 5. Related work
>
> Reflexion adds a self-reflection step and unit test generation step to Self-Debugging - and a self-debugging step to CodeT. So for coding Reflexion has important ideas of both self-debugging and CodeT, yet unlike self-debugging and CodeT, Reflexion is a more general framework that works for diverse tasks.

---

> > ### Comment · Reviewer_V8mX · 2023-08-17
> >
> > Thanks for the experiments and response.
> >
> > (1) Although Reflexion is a general framework, it still seems that on programming tasks "reflexion = self-debugging + CodeT".
> >
> > (2) As a general framework for LLMs, there haven't been enough models considered in the evaluation (like the ones measured by CoT). But I understand that conducting more comprehensive testing requires more time. I hope that the authors can include a more comprehensive set of tests in the future.
> >
> > For the effectiveness of the proposed method, I keep my score and vote for acceptance.

---

### Official Review · Reviewer_k1Ks · 2023-07-06

**Soundness:** 3 good
**Presentation:** 3 good
**Contribution:** 4 excellent
**Rating:** 7
**Confidence:** 3

**Summary:**

The paper introduces Reflexion, a framework for in-context reasoning and decision-making for LMs. It allows the LM to iteratively attempt to solve a task and then reason about the results of its attempts and plan improved courses of action. The authors evaluate Reflexion on a diverse array of tasks (navigating and acting in text environments, question answering, code generation) and report impressive results, in some cases establishing new state-of-the-art. They also compare Reflexion to useful baselines and conduct instructive ablations.

**Strengths:**

1. The paper proposes a sophisticated planning algorithm (composed of short-term and long-term-memory, gating mechanisms and feedback simulators) that can be run (almost) entirely in-context by an LM. I think this approach is very powerful and this paper does a good job showing its promise.
2. The paper is well-written and easy to follow. I like the error analysis paragraph accompanying each section.
3. I appreciate effort put into thorough benchmarking of code generation capabilities, including translating MBPP and HumanEval to Rust and introducing LeetcodeHardGym that prevents tests cases from having been leaked into GPT-4 data.
4. The empirical results obtained by Reflexion are impressive, especially pass@1 for HumanEval.

**Weaknesses:**

1. It would be useful to compare Reflexion to baselines also in terms of inference efficiency. For instance, how many tokens are typically needed for a Reflexion run (vs e.g. ReAct or CoT) to reach an answer in a given task? Is the performance Reflexion bottlenecked by context window size and could we expect it to improve as context windows of LMs increase?
2. Similarly, I am a bit disappointed that authors don’t report any error bars. I understand inference costs, but the authors could at least compute errors bars based on variance across problems (in code generation) or across tasks (in AlfWorld and HotPotQA). Alternatively, I’d rather see a less diverse task array than no error bars.
3. Authors frequently compare Reflexion to human cognition (e.g., “This is akin to how humans iteratively learn  to accomplish complex tasks”, line 31-32; “similar to the way that humans remember fine-grain recent details while also recalling distilled important”, lines 141-143). I think these comparisons are overblown, not supported by cognitive science citations and potentially misleading. For instance, there are numerous dissimilarities between the kind of memory being used in Reflexion and human memory and the topic deserves a more careful discussion that just highlighting superficial similarities in passing.

**Questions:**

1. It's not entirely clear to me which parts of the algorithm happen within the "main" context window of Reflexion and which are encapsulated into separate, local context windows that are then discarded (my understanding is that this is the case, e.g., for unit test generation). I'd appreciate if the authors could make that more clear.
2. How does Reflexion scale (down) with model size? Are GPT-4-level capabilities necessary for it to yield improvements or does it also work with `gpt3.5-turbo` and `text-davinci-003`?
3. How was the error analysis in section 4.1 conducted, i.e. who classified failures as being due hallucination or due to insufficient planning?

**Limitations:**

Certain limitations and societal impacts are discussed. I think it could be good to complement that with a discussion of risks associated with making LMs prompted to be agents act autonomously by allowing them to set their own subgoals and to plan with rich world models (e.g., Perez et al., 2022; Ngo et al., 2023). This could have unknown failure modes if those agents are given unrestricted Internet access and are not supervised.

Perez et al., 2022. Discovering Language Model Behaviors with Model-Written Evaluations

Ngo et al., 2023. The alignment problem from a deep learning perspective

---

> ### Author Rebuttal · Authors · 2023-08-10
>
>
> Thank you for your feedback!
>
> ### 1. Token requirements and context
>
> Reflexion computation scales linearly with respect to the number of episode attempts.
>
> However, with regard to context sizes, Reflexion only keeps last N reflections in the context ($N=1$ for coding and $N=3$ for HotpotQA and AlfWorld), so the context length is not much longer than the base ReAct or CoT context length.
>
> ### 2. Error bars
>
> We were unable to run GPT experiments for many trials due to OpenAI API costs. See the **General Response (2)** for the error bars for StarCoder runs.
> - "at least compute errors bars based on variance across problems (in code generation) or across tasks (in AlfWorld and HotPotQA)": we are not sure what it means, we will be happy to provide any statistics based on existing results.
>
> ### 3. References to human cognition
>
> Thank you for the feedback. We will tone-down lines 31-32 and lines 141-143.
>
> ### 4. Reflexion performance scaled down models
>
> Please see the **General Response (2)**, where we show additional experiments with both stronger and weaker LLMs.
>
> ### 5. Error classification for Alfworld
>
> Error cases were classified by a simple heuristic: If a trajectory contained more than 30 actions, we classified the trial as poorly planned. If a trajectory contained less than 30 actions, but contained 3 consecutive identical action choices, then the trial was classified as a hallucination. This was implemented after our initial experiments as we found that agents were often unable to escape this repetitive action choice loop.

---

> > ### Comment · Reviewer_k1Ks · 2023-08-15
> >
> > Thanks for the detailed response. I especially appreciate experiments with GPT-3.5 series models.
> >
> > I feel my Question 1 from the review wasn't answered. Please let me know if it's not clear.
> >
> > > "at least compute errors bars based on variance across problems (in code generation) or across tasks (in AlfWorld and HotPotQA)": we are not sure what it means, we will be happy to provide any statistics based on existing results.
> >
> > Sure! For instance, HumanEval consists of 164 problems. You could, for a given method, compute pass@1 for each of 164 problems and then average those pass@1 scores across those 164 problems and compute standard errors across 164 problems. This can work even if each pass@1 is estimated based on a single solution attempt (i.e. it's a binary variable; in this case, you can use the closed form of the variance of a binomial distribution).

---

> > > ### Author Response · Authors · 2023-08-16
> > >
> > > **Response to Question 1:**
> > > Reflexion does not have a main context window. There is an implementation, evaluation, and reflection context template that is used. These templates may vary per task to include domain-specific information, such as the permissible action space. The content per step in the loop is specified below:
> > >
> > > Implementation:
> > > - memory (last N items in the list of reflections)
> > > - problem specification
> > > - previous implementation (programming tasks only)
> > >
> > > Evaluation:
> > > - previous implementation
> > > - evaluation instruction
> > >
> > > Reflection:
> > > - previous implementation
> > > - binary success status from evaluation
> > > - reflection instruction
> > >
> > > **Error bars**
> > > Thank you for the additional information! Initially, we did not do trial-level variance estimates over question success because HumanEval problems are not equal in difficulty. However, here is the standard error for HumanEval (python):
> > > SE = sqrt((0.91)(1-0.91)/164) = 0.022
> > >
> > > Thank you very much for your clarification!

---

> > > > ### Comment · Reviewer_k1Ks · 2023-08-16
> > > >
> > > > Thanks for the clarifications, they are helpful.
> > > >
> > > > > Initially, we did not do trial-level variance estimates over question success because HumanEval problems are not equal in difficulty.
> > > >
> > > > I see your point. I think standard error still provides a useful measure of result's variability (and how significant is the gap between Reflexion and other methods) and I think the paper would benefit from reporting results with standard errors computed this way. But I won't press on that if the Authors disagree.

---

> > > > > ### Author Response · Authors · 2023-08-17
> > > > > **Yes**
> > > > >
> > > > > Thanks for your prompt reply, and finding our clarification helpful!
> > > > >
> > > > > Yes, we agree reporting standard error can be useful. We will report it in revision.

---

### Author Rebuttal · Authors · 2023-08-10


We appreciate all of the great feedback from our reviewers.

### 1. Related Work, Ablations, and Baselines

Reflexion is a general framework for reasoning, decision-making, and programming problems. A typical Reflexion loop consists of a generation, evaluation, and feedback function. Thus, the setup for the individual components are different, but similar across all tasks. To demonstrate the requirement for self-reflection, we precisely removed individual components of the loop to evaluate accuracy. On HotPotQA (Figure 4C), we show that having simple episodic memory (full past trajectories) in the context was not as effective as self-reflection instruction. On HumanEval Rust (Table 3), we show that removing the self-reflection step results in the same performance as the baseline accuracy. Many other related works such as Self-Debugging and CodeT lack at least one of: self-written tests, natural language reflections, or generalizability.

### 2. Different LLMs and Additional Experiments

- HotpotQA: We used `text-davinci-003` for the Actor and Self-Reflector.
- Alfworld: We used `text-davinci-003` for the Actor, Evaluator, and Self-Reflector.
- Programming tasks: We used `gpt-4` for the Actor, Evaluator, and Self-Reflector.
- Additional experiment #1:
    - We ran HumanEval Python with `starchat-beta`, an open-source model that was pre-trained on a large open-source dataset of permissively licensed code, `The Stack`, as the Actor, Evaluator, and Reflector.
    - We found that low reflection generation quality prevented improvement in performance.
- Additional experiment #2:
    - We ran HotPotQA with `gpt-3.5-turbo` and `gpt-4` as the Actor, Evaluator, and Reflector to test Reflexion performance using stronger models.
    - It was not surprising that we found that Reflexion agents were able to learn with stronger models.

We ran two new experiments to evaluate Reflexion using `starchat-beta` on HumanEval, and `gpt-4` and `gpt-3.5-turbo` on HotPotQA.

|          |  HumanEval Pass@1  |
| -------- | ------- |
| Baseline | 0.26 (avg over 8 runs); SE = 0.00481 |
| Reflexion | 0.26 (avg over 8 runs); SE = 0.00305 |

StarChat is qualitatively and quantitatively not suitable for running a Reflexion loop. For example, given the following erroneous implementation:

```python
# Adds two numbers
def add(x, y):
  if x == 0:
    return 0
  else:
    return x + y
```

And feedback from the unit test suite:

```python
# passing tests
assert add(1, 2) == 3
assert add(3, 0) == 3

# failing tests
assert add(0, 2) == 2 # Output: 0
```

StarChat generates the following self-reflection which is not good:


    In this case, it appears as though there may have been an issue with scope or variable shadowing occurring within your function definition for add(). Specifically, in line 4 of your provided code snippet, where you define the local variable x with the value of 0, but then later reference the global variable x (which has no initial value assigned), it could potentially cause issues depending on how the rest of your function is written.

    To resolve this error, I would recommend explicitly defining a new variable called zero which takes on the value of 0 at the beginning of your function, like so:

    ```python
    def add(x,y):
        zero = 0
        if x==0:
            return zero
        else:
            return x+y
    ```

    This should ensure that the intended behavior of returning zero when x=0 remains intact while also preventing potential confusion around scoping/shadowing. Let me know if you need anything further!

On the other hand, we scaled-up the models for Reflexion evaluation on HotPotQA. We found that Reflexion enabled CoT and ReAct agents to improve performance.


| Model | Baseline | Reflexion |
| -------- | ------- | ------- |
| CoT (GT) + text-davinci-003 | 0.60 | 0.77 |
| CoT (GT) + gpt-3.5-turbo | 0.57 | 0.71 |
| CoT (GT) + gpt-4 | 0.68 | 0.80 |
| ReAct + text-davinci-003 | 0.3 | 0.55 |
| ReAct + gpt-3.5-turbo | 0.26 | 0.38 |
| ReAct + gpt-4 | 0.39 | 0.51 |

We believe that Reflexion will be strengthened as LLMs improve given that the ability to effectively self-reflect is crucial to performance.

---

### Author Response · Authors · 2023-08-17
**Thanks for Responses, and Look Forward to Discussions**

We want to thank Reviewers k1Ks, V8mX, 6XLB for responding to our rebuttal, and we look forward to engaging in discussions with Reviewers BKdQ, U48e, to see if their concerns are addressed, and address any potential remaining concerns in the final 4 days. Thanks in advance!

---

### Decision · Program_Chairs · 2023-09-21

**Decision:**

Accept (poster)

**Comment:**

In this paper, the authors proposed to treat LLM as an RL agent through linguistic feedback. Specifically, LLM agents reflect verbally on feedback signals and maintain a memory buffer to facilitate decision-making. The approach incorporates different types of feedback signals and can be applied to different types of tasks e.g. coding, language reasoning, and decision-making.

All the authors appreciated the novelty of the proposed approach, especially in the short-term and long-term memory mechanism and natural feedback models, as well as the comprehensive experiments including a new benchmark LeetcodeHardGym. The results are significantly improved from baseline models, obtaining SOTA results in different types of tasks. The reviewers also suggested several ways to improve the paper, including a discussion of human cognition, and additional results from application to other LLMs besides OpenAI models. I hope the authors can incorporate some of these changes to improve their final paper.